# Variational Bayesian Monte Carlo

**Luigi Acerbi**[*]
Department of Basic Neuroscience
University of Geneva
`luigi.acerbi@unige.ch`

## Abstract

Many probabilistic models of interest in scientific computing and machine learning have expensive, black-box likelihoods that prevent the application of standard techniques for Bayesian inference, such as MCMC, which would require access to the gradient or a large number of likelihood evaluations. We introduce here a novel sample-efficient inference framework, Variational Bayesian Monte Carlo (VBMC). VBMC combines variational inference with Gaussian-process based, active-sampling Bayesian quadrature, using the latter to efficiently approximate the intractable integral in the variational objective. Our method produces both a nonparametric approximation of the posterior distribution and an approximate lower bound of the model evidence, useful for model selection. We demonstrate VBMC both on several synthetic likelihoods and on a neuronal model with data from real neurons. Across all tested problems and dimensions (up to $D = 10$), VBMC performs consistently well in reconstructing the posterior and the model evidence with a limited budget of likelihood evaluations, unlike other methods that work only in very low dimensions. Our framework shows great promise as a novel tool for posterior and model inference with expensive, black-box likelihoods.

## 1 Introduction

In many scientific, engineering, and machine learning domains, such as in computational neuroscience and big data, complex black-box computational models are routinely used to estimate model parameters and compare hypotheses instantiated by different models. Bayesian inference allows us to do so in a principled way that accounts for parameter and model uncertainty by computing the posterior distribution over parameters and the model evidence, also known as marginal likelihood or Bayes factor. However, Bayesian inference is generally analytically intractable, and the statistical tools of approximate inference, such as Markov Chain Monte Carlo (MCMC) or variational inference, generally require knowledge about the model (e.g., access to the gradients) and/or a large number of model evaluations. Both of these requirements cannot be met by black-box probabilistic models with computationally expensive likelihoods, precluding the application of standard Bayesian techniques of parameter and model uncertainty quantification to domains that would most need them.

Given a dataset $\mathcal{D}$ and model parameters $\boldsymbol{x} \in \mathbb{R}^D$, here we consider the problem of computing both the *posterior* $p(\boldsymbol{x}|\mathcal{D})$ and the *marginal likelihood* (or model evidence) $p(\mathcal{D})$, defined as, respectively,

$$p(\boldsymbol{x}|\mathcal{D}) = \frac{p(\mathcal{D}|\boldsymbol{x})p(\boldsymbol{x})}{p(\mathcal{D})} \qquad \text{and} \qquad p(\mathcal{D}) = \int p(\mathcal{D}|\boldsymbol{x})p(\boldsymbol{x})d\boldsymbol{x}, \qquad (1)$$

where $p(\mathcal{D}|\boldsymbol{x})$ is the likelihood of the model of interest and $p(\boldsymbol{x})$ is the prior over parameters. Crucially, we consider the case in which $p(\mathcal{D}|\boldsymbol{x})$ is a black-box, expensive function for which we have a limited budget of function evaluations (of the order of few hundreds).

A promising approach to deal with such computational constraints consists of building a probabilistic model-based approximation of the function of interest, for example via Gaussian processes (GP)

---

[*]Website: `luigiacerbi.com`. Alternative e-mail: `luigi.acerbi@gmail.com`.

[1]. This statistical surrogate can be used in lieu of the original, expensive function, allowing faster computations. Moreover, uncertainty in the surrogate can be used to actively guide sampling of the original function to obtain a better approximation in regions of interest for the application at hand. This approach has been extremely successful in Bayesian optimization [2, 3, 4, 5, 6] and in Bayesian quadrature for the computation of intractable integrals [7, 8].

In particular, recent works have applied GP-based Bayesian quadrature to the estimation of the marginal likelihood [8, 9, 10, 11], and GP surrogates to build approximations of the posterior [12, 13]. However, none of the existing approaches deals simultaneously with posterior and model inference. Moreover, it is unclear how these approximate methods would deal with likelihoods with realistic properties, such as medium dimensionality (up to $D \sim 10$), mild multi-modality, heavy tails, and parameters that exhibit strong correlations—all common issues of real-world applications.

In this work, we introduce Variational Bayesian Monte Carlo (VBMC), a novel approximate inference framework that combines variational inference and active-sampling Bayesian quadrature via GP surrogates.[1] Our method affords simultaneous approximation of the posterior and of the model evidence in a sample-efficient manner. We demonstrate the robustness of our approach by testing VBMC and other inference algorithms on a variety of synthetic likelihoods with realistic, challenging properties. We also apply our method to a real problem in computational neuroscience, by fitting a model of neuronal selectivity in visual cortex [14]. Among the tested methods, VBMC is the only one with consistently good performance across problems, showing promise as a novel tool for posterior and model inference with expensive likelihoods in scientific computing and machine learning.

## 2 Theoretical background

### 2.1 Variational inference

Variational Bayes is an approximate inference method whereby the posterior $p(\boldsymbol{x}|\mathcal{D})$ is approximated by a simpler distribution $q(\boldsymbol{x}) \equiv q_{\boldsymbol{\phi}}(\boldsymbol{x})$ that usually belongs to a parametric family [15, 16]. The goal of variational inference is to find the variational parameters $\boldsymbol{\phi}$ for which the variational posterior $q_{\boldsymbol{\phi}}$ "best" approximates the true posterior. In variational methods, the mismatch between the two distributions is quantified by the Kullback-Leibler (KL) divergence,

$$\text{KL}\left[q_{\boldsymbol{\phi}}(\boldsymbol{x})||p(\boldsymbol{x}|\mathcal{D})\right] = \mathbb{E}_{\boldsymbol{\phi}}\left[\log \frac{q_{\boldsymbol{\phi}}(\boldsymbol{x})}{p(\boldsymbol{x}|\mathcal{D})}\right], \tag{2}$$

where we adopted the compact notation $\mathbb{E}_{\boldsymbol{\phi}} \equiv \mathbb{E}_{q_{\boldsymbol{\phi}}}$. Inference is then reduced to an optimization problem, that is finding the variational parameter vector $\boldsymbol{\phi}$ that minimizes Eq. 2. We rewrite Eq. 2 as

$$\log p(\mathcal{D}) = \mathcal{F}[q_{\boldsymbol{\phi}}] + \text{KL}\left[q_{\boldsymbol{\phi}}(\boldsymbol{x})||p(\boldsymbol{x}|\mathcal{D})\right], \tag{3}$$

where

$$\mathcal{F}\left[q_{\boldsymbol{\phi}}\right] = \mathbb{E}_{\boldsymbol{\phi}}\left[\log \frac{p(\mathcal{D}|\boldsymbol{x})p(\boldsymbol{x})}{q_{\boldsymbol{\phi}}(\boldsymbol{x})}\right] = \mathbb{E}_{\boldsymbol{\phi}}\left[f(\boldsymbol{x})\right] + \mathcal{H}[q_{\boldsymbol{\phi}}(\boldsymbol{x})] \tag{4}$$

is the negative free energy, or *evidence lower bound* (ELBO). Here $f(\boldsymbol{x}) \equiv \log p(\mathcal{D}|\boldsymbol{x})p(\boldsymbol{x}) = \log p(\mathcal{D}, \boldsymbol{x})$ is the log joint probability and $\mathcal{H}[q]$ is the entropy of $q$. Note that since the KL divergence is always non-negative, from Eq. 3 we have $\mathcal{F}[q] \leq \log p(\mathcal{D})$, with equality holding if $q(\boldsymbol{x}) \equiv p(\boldsymbol{x}|\mathcal{D})$. Thus, maximization of the variational objective, Eq. 4, is equivalent to minimization of the KL divergence, and produces both an approximation of the posterior $q_{\boldsymbol{\phi}}$ and a lower bound on the marginal likelihood, which can be used as a metric for model selection.

Normally, $q$ is chosen to belong to a family (e.g., a factorized posterior, or mean field) such that the expected log joint in Eq. 4 and the entropy can be computed analytically, possibly providing closed-form equations for a coordinate ascent algorithm. Here, we assume that $f(\boldsymbol{x})$, like many computational models of interest, is an expensive black-box function, which prevents a direct computation of Eq. 4 analytically or via simple numerical integration.

### 2.2 Bayesian quadrature

Bayesian quadrature, also known as Bayesian Monte Carlo, is a means to obtain Bayesian estimates of the mean and variance of non-analytical integrals of the form $\langle f \rangle = \int f(\boldsymbol{x})\pi(\boldsymbol{x})d\boldsymbol{x}$, defined on

a domain $\mathcal{X} = \mathbb{R}^D$ [7, 8]. Here, $f$ is a function of interest and $\pi$ a known probability distribution. Typically, a Gaussian Process (GP) prior is specified for $f(\boldsymbol{x})$.

**Gaussian processes**   GPs are a flexible class of models for specifying prior distributions over unknown functions $f : \mathcal{X} \subseteq \mathbb{R}^D \to \mathbb{R}$ [1]. GPs are defined by a mean function $m : \mathcal{X} \to \mathbb{R}$ and a positive definite covariance, or kernel function $\kappa : \mathcal{X} \times \mathcal{X} \to \mathbb{R}$. In Bayesian quadrature, a common choice is the Gaussian kernel $\kappa(\boldsymbol{x}, \boldsymbol{x}') = \sigma_f^2 \mathcal{N}(\boldsymbol{x}; \boldsymbol{x}', \boldsymbol{\Sigma}_\ell)$, with $\boldsymbol{\Sigma}_\ell = \mathrm{diag}[\ell^{(1)^2}, \ldots, \ell^{(D)^2}]$, where $\sigma_f$ is the output length scale and $\boldsymbol{\ell}$ is the vector of input length scales. Conditioned on training inputs $\mathbf{X} = \{\boldsymbol{x}_1, \ldots, \boldsymbol{x}_n\}$ and associated function values $\boldsymbol{y} = f(\mathbf{X})$, the GP posterior will have latent posterior conditional mean $\overline{f}_{\boldsymbol{\Xi}}(\boldsymbol{x}) \equiv \overline{f}(\boldsymbol{x}; \boldsymbol{\Xi}, \boldsymbol{\psi})$ and covariance $C_{\boldsymbol{\Xi}}(\boldsymbol{x}, \boldsymbol{x}') \equiv C(\boldsymbol{x}, \boldsymbol{x}'; \boldsymbol{\Xi}, \boldsymbol{\psi})$ in closed form (see [1]), where $\boldsymbol{\Xi} = \{\mathbf{X}, \boldsymbol{y}\}$ is the set of training function data for the GP and $\boldsymbol{\psi}$ is a hyperparameter vector for the GP mean, covariance, and likelihood.

**Bayesian integration**   Since integration is a linear operator, if $f$ is a GP, the posterior mean and variance of the integral $\int f(\boldsymbol{x})\pi(\boldsymbol{x})d\boldsymbol{x}$ are [8]

$$\mathbb{E}_{f|\boldsymbol{\Xi}}[\langle f \rangle] = \int \overline{f}_{\boldsymbol{\Xi}}(\boldsymbol{x})\pi(\boldsymbol{x})d\boldsymbol{x}, \qquad \mathbb{V}_{f|\boldsymbol{\Xi}}[\langle f \rangle] = \int\int C_{\boldsymbol{\Xi}}(\boldsymbol{x}, \boldsymbol{x}')\pi(\boldsymbol{x})d\boldsymbol{x}\pi(\boldsymbol{x}')d\boldsymbol{x}'. \tag{5}$$

Crucially, if $f$ has a Gaussian kernel and $\pi$ is a Gaussian or mixture of Gaussians (among other functional forms), the integrals in Eq. 5 can be computed analytically.

**Active sampling**   For a given budget of samples $n_{\max}$, a smart choice of the input samples $\mathbf{X}$ would aim to minimize the posterior variance of the final integral (Eq. 5) [11]. Interestingly, for a standard GP and fixed GP hyperparameters $\boldsymbol{\psi}$, the optimal variance-minimizing design does not depend on the function values at $\mathbf{X}$, thereby allowing precomputation of the optimal design. However, if the GP hyperparameters are updated online, or the GP is warped (e.g., via a log transform [9] or a square-root transform [10]), the variance of the posterior will depend on the function values obtained so far, and an active sampling strategy is desirable. The *acquisition function* $a : \mathcal{X} \to \mathbb{R}$ determines which point in $\mathcal{X}$ should be evaluated next via a proxy optimization $\boldsymbol{x}_{\text{next}} = \mathrm{argmax}_{\boldsymbol{x}} a(\boldsymbol{x})$. Examples of acquisition functions for Bayesian quadrature include the *expected entropy*, which minimizes the expected entropy of the integral after adding $\boldsymbol{x}$ to the training set [9], and the much faster to compute *uncertainty sampling* strategy, which maximizes the variance of the *integrand* at $\boldsymbol{x}$ [10].

# 3   Variational Bayesian Monte Carlo (VBMC)

We introduce here Variational Bayesian Monte Carlo (VBMC), a sample-efficient inference method that combines variational Bayes and Bayesian quadrature, particularly useful for models with (moderately) expensive likelihoods. The main steps of VBMC are described in Algorithm 1, and an example run of VBMC on a nontrivial 2-D target density is shown in Fig. 1.

**VBMC in a nutshell**   In each iteration $t$, the algorithm: (1) sequentially samples a batch of 'promising' new points that maximize a given acquisition function, and evaluates the (expensive) log joint $f$ at each of them; (2) trains a GP model of the log joint $f$, given the training set $\boldsymbol{\Xi}_t = \{\mathbf{X}_t, \boldsymbol{y}_t\}$ of points evaluated so far; (3) updates the variational posterior approximation, indexed by $\boldsymbol{\phi}_t$, by optimizing the ELBO. This loop repeats until the budget of function evaluations is exhausted, or some other termination criterion is met (e.g., based on the stability of the found solution). VBMC includes an initial *warm-up* stage to avoid spending computations in regions of low posterior probability mass (see Section 3.5). In the following sections, we describe various features of VBMC.

## 3.1   Variational posterior

We choose for the variational posterior $q(\boldsymbol{x})$ a flexible "nonparametric" family, a mixture of $K$ Gaussians with shared covariances, modulo a scaling factor,

$$q(\boldsymbol{x}) \equiv q_{\boldsymbol{\phi}}(\boldsymbol{x}) = \sum_{k=1}^{K} w_k \mathcal{N}\left(\boldsymbol{x}; \boldsymbol{\mu}_k, \sigma_k^2 \boldsymbol{\Sigma}\right), \tag{6}$$

where $w_k$, $\boldsymbol{\mu}_k$, and $\sigma_k$ are, respectively, the mixture weight, mean, and scale of the $k$-th component, and $\boldsymbol{\Sigma}$ is a covariance matrix common to all elements of the mixture. In the following, we assume

**Algorithm 1** Variational Bayesian Monte Carlo

---

**Input:** target log joint $f$, starting point $\boldsymbol{x_0}$, plausible bounds PLB, PUB, additional `options`
1: **Initialization:** $t \leftarrow 0$, initialize variational posterior $\phi_0$, STOPSAMPLING $\leftarrow$ `false`
2: **repeat**
3:     $t \leftarrow t + 1$
4:     **if** $t \triangleq 1$ **then**                                              ▷ Initial design, Section 3.5
5:         Evaluate $y_0 \leftarrow f(\boldsymbol{x}_0)$ and add $(\boldsymbol{x}_0, y_0)$ to the training set $\boldsymbol{\Xi}$
6:         **for** $2 \ldots n_{\text{init}}$ **do**
7:             Sample a new point $\boldsymbol{x}_{\text{new}} \leftarrow \text{Uniform}[\texttt{PLB}, \texttt{PUB}]$
8:             Evaluate $y_{\text{new}} \leftarrow f(\boldsymbol{x}_{\text{new}})$ and add $(\boldsymbol{x}_{\text{new}}, y_{\text{new}})$ to the training set $\boldsymbol{\Xi}$
9:     **else**
10:        **for** $1 \ldots n_{\text{active}}$ **do**                                 ▷ Active sampling, Section 3.3
11:            Actively sample a new point $\boldsymbol{x}_{\text{new}} \leftarrow \text{argmax}_{\boldsymbol{x}} a(\boldsymbol{x})$
12:            Evaluate $y_{\text{new}} \leftarrow f(\boldsymbol{x}_{\text{new}})$ and add $(\boldsymbol{x}_{\text{new}}, y_{\text{new}})$ to the training set $\boldsymbol{\Xi}$
13:            **for each** $\boldsymbol{\psi}_1, \ldots, \boldsymbol{\psi}_{n_{\text{gp}}}$, perform rank-1 update of the GP posterior
14:        **if not** STOPSAMPLING **then**                        ▷ GP hyperparameter training, Section 3.4
15:            $\{\boldsymbol{\psi}_1, \ldots, \boldsymbol{\psi}_{n_{\text{gp}}}\} \leftarrow$ Sample GP hyperparameters
16:        **else**
17:            $\boldsymbol{\psi}_1 \leftarrow$ Optimize GP hyperparameters
18:        $K_t \leftarrow$ Update number of variational components                        ▷ Section 3.6
19:        $\phi_t \leftarrow$ Optimize ELBO via stochastic gradient descent                ▷ Section 3.2
20:        Evaluate whether to STOPSAMPLING and other TERMINATIONCRITERIA
21: **until** `fevals` > `MaxFunEvals` **or** TERMINATIONCRITERIA   ▷ Stopping criteria, Section 3.7
22: **return** variational posterior $\phi_t$, $\mathbb{E}\left[\text{ELBO}\right]$, $\sqrt{\mathbb{V}\left[\text{ELBO}\right]}$

---

a diagonal matrix $\boldsymbol{\Sigma} \equiv \text{diag}[\lambda^{(1)2}, \ldots, \lambda^{(D)2}]$. The variational posterior for a given number of mixture components $K$ is parameterized by $\boldsymbol{\phi} \equiv (w_1, \ldots, w_K, \boldsymbol{\mu}_1, \ldots, \boldsymbol{\mu}_K, \sigma_1, \ldots, \sigma_K, \boldsymbol{\lambda})$, which has $K(D + 2) + D$ parameters. The number of components $K$ is set adaptively (see Section 3.6).

### 3.2 The evidence lower bound

We approximate the ELBO (Eq. 4) in two ways. First, we approximate the log joint probability $f$ with a GP with a squared exponential (rescaled Gaussian) kernel, a Gaussian likelihood with observation noise $\sigma_{\text{obs}} > 0$ (for numerical stability [17]), and a *negative quadratic* mean function, defined as

$$m(\boldsymbol{x}) = m_0 - \frac{1}{2} \sum_{i=1}^{D} \frac{\left(x^{(i)} - x_{\text{m}}^{(i)}\right)^2}{\omega^{(i)2}}, \tag{7}$$

where $m_0$ is the maximum value of the mean, $\boldsymbol{x}_{\text{m}}$ is the location of the maximum, and $\boldsymbol{\omega}$ is a vector of length scales. This mean function, unlike for example a constant mean, ensures that the posterior GP predictive mean $\overline{f}$ is a proper log probability distribution (that is, it is integrable when exponentiated). Crucially, our choice of variational family (Eq. 6) and kernel, likelihood and mean function of the GP affords an analytical computation of the posterior mean and variance of the expected log joint $\mathbb{E}_{\phi}[f]$ (using Eq. 5), and of their gradients (see Supplementary Material for details). Second, we approximate the entropy of the variational posterior, $\mathcal{H}[q_{\phi}]$, via simple Monte Carlo sampling, and we propagate its gradient through the samples via the reparametrization trick [18, 19].[2] Armed with expressions for the mean expected log joint, the entropy, and their gradients, we can efficiently optimize the (negative) mean ELBO via stochastic gradient descent [21].

**Evidence lower confidence bound**  We define the *evidence lower confidence bound* (ELCBO) as

$$\text{ELCBO}(\phi, f) = \mathbb{E}_{f|\boldsymbol{\Xi}}\left[\mathbb{E}_{\phi}[f]\right] + \mathcal{H}[q_{\phi}] - \beta_{\text{LCB}}\sqrt{\mathbb{V}_{f|\boldsymbol{\Xi}}\left[\mathbb{E}_{\phi}[f]\right]} \tag{8}$$

where the first two terms are the ELBO (Eq. 4) estimated via Bayesian quadrature, and the last term is the uncertainty in the computation of the expected log joint multiplied by a risk-sensitivity parameter

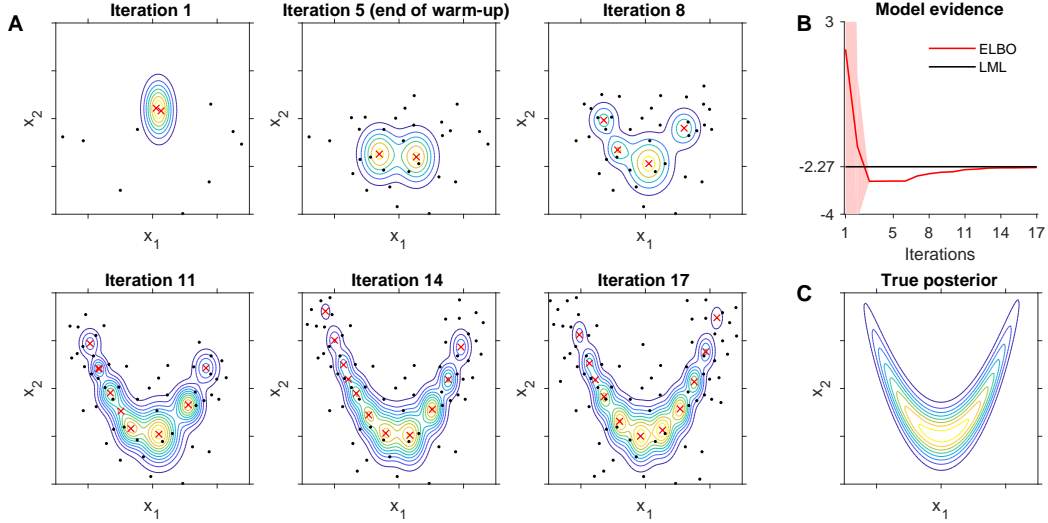

Figure 1: **Example run of VBMC on a 2-D pdf. A.** Contour plots of the variational posterior at different iterations of the algorithm. Red crosses indicate the centers of the variational mixture components, black dots are the training samples. **B.** ELBO as a function of iteration. Shaded area is 95% CI of the ELBO in the current iteration as per the Bayesian quadrature approximation (*not* the error wrt ground truth). The black line is the true log marginal likelihood (LML). **C.** True target pdf.

$\beta_{\mathrm{LCB}}$ (we set $\beta_{\mathrm{LCB}} = 3$ unless specified otherwise). Eq. 8 establishes a probabilistic lower bound on the ELBO, used to assess the improvement of the variational solution (see following sections).

### 3.3 Active sampling

In VBMC, we are performing active sampling to compute a *sequence* of integrals $\mathbb{E}_{\phi_1}[f], \mathbb{E}_{\phi_2}[f], \ldots, \mathbb{E}_{\phi_T}[f]$, across iterations $1, \ldots, T$ such that (1) the sequence of variational parameters $\phi_t$ converges to the variational posterior that minimizes the KL divergence with the true posterior, and (2) we have minimum variance on our final estimate of the ELBO. Note how this differs from active sampling in simple Bayesian quadrature, for which we only care about minimizing the variance of a single fixed integral. The ideal acquisition function for VBMC will correctly balance exploration of uncertain regions and exploitation of regions with high probability mass to ensure a fast convergence of the variational posterior as closely as possible to the ground truth.

We describe here two acquisition functions for VBMC based on uncertainty sampling. Let $V_{\boldsymbol{\Xi}}(\boldsymbol{x}) \equiv C_{\boldsymbol{\Xi}}(\boldsymbol{x}, \boldsymbol{x})$ be the posterior GP variance at $\boldsymbol{x}$ given the current training set $\boldsymbol{\Xi}$. 'Vanilla' uncertainty sampling for $\mathbb{E}_{\phi}[f]$ is $a_{\mathrm{us}}(\boldsymbol{x}) = V_{\boldsymbol{\Xi}}(\boldsymbol{x})q_{\phi}(\boldsymbol{x})^2$, where $q_{\phi}$ is the current variational posterior. Since $a_{\mathrm{us}}$ only maximizes the variance of the integrand under the *current* variational parameters, we expect it to be lacking in exploration. To promote exploration, we introduce *prospective uncertainty sampling*,

$$a_{\mathrm{pro}}(\boldsymbol{x}) = V_{\boldsymbol{\Xi}}(\boldsymbol{x})q_{\phi}(\boldsymbol{x}) \exp\left(\overline{f}_{\boldsymbol{\Xi}}(\boldsymbol{x})\right), \tag{9}$$

where $\overline{f}_{\boldsymbol{\Xi}}$ is the GP posterior predictive mean. $a_{\mathrm{pro}}$ aims at reducing uncertainty of the variational objective both for the current posterior and at prospective locations where the variational posterior might move to in the future, if not already there (high GP posterior mean). The variational posterior in $a_{\mathrm{pro}}$ acts as a regularizer, preventing active sampling from following too eagerly fluctuations of the GP mean. For numerical stability of the GP, we include in all acquisition functions a regularization factor to prevent selection of points too close to existing training points (see Supplementary Material).

At the beginning of each iteration after the first, VBMC actively samples $n_{\mathrm{active}}$ points ($n_{\mathrm{active}} = 5$ by default in this work). We select each point sequentially, by optimizing the chosen acquisition function via CMA-ES [22], and apply fast rank-one updates of the GP posterior after each acquisition.

### 3.4 Adaptive treatment of GP hyperparameters

The GP model in VBMC has $3D + 3$ hyperparameters, $\boldsymbol{\psi} = (\boldsymbol{\ell}, \sigma_f, \sigma_{\mathrm{obs}}, m_0, \boldsymbol{x}_{\mathrm{m}}, \boldsymbol{\omega})$. We impose an empirical Bayes prior on the GP hyperparameters based on the current training set (see Supplementary

Material), and we sample from the posterior over hyperparameters via slice sampling [23]. In each iteration, we collect $n_{\mathrm{gp}} = \mathrm{round}(80/\sqrt{n})$ samples, where $n$ is the size of the current GP training set, with the rationale that we require less samples as the posterior over hyperparameters becomes narrower due to more observations. Given samples $\{\boldsymbol{\psi}\} \equiv \{\boldsymbol{\psi}_1, \ldots, \boldsymbol{\psi}_{n_{\mathrm{gp}}}\}$, and a random variable $\chi$ that depends on $\boldsymbol{\psi}$, we compute the expected mean and variance of $\chi$ as

$$\mathbb{E}\left[\chi | \{\boldsymbol{\psi}\}\right] = \frac{1}{n_{\mathrm{gp}}} \sum_{j=1}^{n_{\mathrm{gp}}} \mathbb{E}\left[\chi | \boldsymbol{\psi}_j\right], \quad \mathbb{V}\left[\chi | \{\boldsymbol{\psi}\}\right] = \frac{1}{n_{\mathrm{gp}}} \sum_{j=1}^{n_{\mathrm{gp}}} \mathbb{V}\left[\chi | \boldsymbol{\psi}_j\right] + \mathrm{Var}\left[\{\mathbb{E}\left[\chi | \boldsymbol{\psi}_j\right]\}_{j=1}^{n_{\mathrm{gp}}}\right], \quad (10)$$

where $\mathrm{Var}[\cdot]$ is the sample variance. We use Eq. 10 to compute the GP posterior predictive mean and variances for the acquisition function, and to marginalize the expected log joint over hyperparameters.

The algorithm adaptively switches to a faster maximum-a-posteriori (MAP) estimation of the hyperparameters (via gradient-based optimization) when the additional variability of the expected log joint brought by multiple samples falls below a threshold for several iterations, a signal that sampling is bringing little advantage to the precision of the computation.

### 3.5 Initialization and warm-up

The algorithm is initialized by providing a starting point $\boldsymbol{x}_0$ (ideally, in a region of high posterior probability mass) and vectors of *plausible* lower/upper bounds PLB, PUB, that identify a region of high posterior probability mass in parameter space. In the absence of other information, we obtained good results with plausible bounds containing the peak of prior mass in each coordinate dimension, such as the top $\sim 0.68$ probability region (that is, mean $\pm 1$ SD for a Gaussian prior). The initial design consists of the provided starting point(s) $\boldsymbol{x}_0$ and additional points generated uniformly at random inside the plausible box, for a total of $n_{\mathrm{init}} = 10$ points. The plausible box also sets the reference scale for each variable, and in future work might inform other aspects of the algorithm [6]. The VBMC algorithm works in an unconstrained space ($\boldsymbol{x} \in \mathbb{R}^D$), but bound constraints to the variables can be easily handled via a nonlinear remapping of the input space, with an appropriate Jacobian correction of the log probability density [24] (see Section 4.2 and Supplementary Material).[3]

**Warm-up** We initialize the variational posterior with $K = 2$ components in the vicinity of $\boldsymbol{x}_0$, and with small values of $\sigma_1, \sigma_2$, and $\boldsymbol{\lambda}$ (relative to the width of the plausible box). The algorithm starts in *warm-up* mode, during which VBMC tries to quickly improve the ELBO by moving to regions with higher posterior probability. During warm-up, $K$ is clamped to only two components with $w_1 \equiv w_2 = 1/2$, and we collect a maximum of $n_{\mathrm{gp}} = 8$ hyperparameter samples. Warm-up ends when the ELCBO (Eq. 8) shows an improvement of less than 1 for three consecutive iterations, suggesting that the variational solution has started to stabilize. At the end of warm-up, we *trim* the training set by removing points whose value of the log joint probability $y$ is more than $10 \cdot D$ points lower than the maximum value $y_{\mathrm{max}}$ observed so far. While not necessary in theory, we found that trimming generally increases the stability of the GP approximation, especially when VBMC is initialized in a region of very low probability under the true posterior. To allow the variational posterior to adapt, we do not actively sample new points in the first iteration after the end of warm-up.

### 3.6 Adaptive number of variational mixture components

After warm-up, we add and remove variational components following a simple set of rules.

**Adding components** We define the current variational solution as *improving* if the ELCBO of the last iteration is higher than the ELCBO in the past few iterations ($n_{\mathrm{recent}} = 4$). In each iteration, we increment the number of components $K$ by 1 if the solution is improving and no mixture component was *pruned* in the last iteration (see below). To speed up adaptation of the variational solution to a complex true posterior when the algorithm has nearly converged, we further add two extra components if the solution is *stable* (see below) and no component was recently pruned. Each new component is created by splitting and jittering a randomly chosen existing component. We set a maximum number of components $K_{\mathrm{max}} = n^{2/3}$, where $n$ is the size of the current training set $\boldsymbol{\Xi}$.

**Removing components** At the end of each variational optimization, we consider as a candidate for *pruning* a random mixture component $k$ with mixture weight $w_k < w_{\mathrm{min}}$. We recompute the ELCBO

without the selected component (normalizing the remaining weights). If the 'pruned' ELCBO differs from the original ELCBO less than $\varepsilon$, we remove the selected component. We iterate the process through all components with weights below threshold. For VBMC we set $w_{\min} = 0.01$ and $\varepsilon = 0.01$.

### 3.7 Termination criteria

At the end of each iteration, we assign a *reliability index* $\rho(t) \geq 0$ to the current variational solution based on the following features: change in ELBO between the current and the previous iteration; estimated variance of the ELBO; KL divergence between the current and previous variational posterior (see Supplementary Material for details). By construction, a $\rho(t) \lesssim 1$ is suggestive of a *stable* solution. The algorithm terminates when obtaining a stable solution for $n_{\text{stable}} = 8$ iterations (with at most one non-stable iteration in-between), or when reaching a maximum number $n_{\max}$ of function evaluations. The algorithm returns the estimate of the mean and standard deviation of the ELBO (a lower bound on the marginal likelihood), and the variational posterior, from which we can cheaply draw samples for estimating distribution moments, marginals, and other properties of the posterior. If the algorithm terminates before achieving long-term stability, it warns the user and returns a recent solution with the best ELCBO, using a conservative $\beta_{\text{LCB}} = 5$.

## 4 Experiments

We tested VBMC and other common inference algorithms on several artificial and real problems consisting of a target likelihood and an associated prior. The goal of inference consists of approximating the posterior distribution and the log marginal likelihood (LML) with a fixed budget of likelihood evaluations, assumed to be (moderately) expensive.

**Algorithms** We tested VBMC with the 'vanilla' uncertainty sampling acquisition function $a_{\text{us}}$ (VBMC-U) and with prospective uncertainty sampling, $a_{\text{pro}}$ (VBMC-P). We also tested simple Monte Carlo (SMC), annealed importance sampling (AIS), the original Bayesian Monte Carlo (BMC), doubly-Bayesian quadrature (BBQ [9])[4], and warped sequential active Bayesian integration (WSABI, both in its linearized and moment-matching variants, WSABI-L and WSABI-M [10]). For the basic setup of these methods, we follow [10]. Most of these algorithms only compute an approximation of the marginal likelihood based on a set of sampled points, but do not directly compute a posterior distribution. We obtain a posterior by training a GP model (equal to the one used by VBMC) on the log joint evaluated at the sampled points, and then drawing $2 \cdot 10^4$ MCMC samples from the GP posterior predictive mean via parallel slice sampling [23, 25]. We also tested two methods for posterior estimation via GP surrogates, BAPE [12] and AGP [13]. Since these methods only compute an approximate posterior, we obtain a crude estimate of the log normalization constant (the LML) as the average difference between the log of the approximate posterior and the evaluated log joint at the top 20% points in terms of posterior density. For all algorithms, we use default settings, allowing only changes based on knowledge of the mean and (diagonal) covariance of the provided prior.

**Procedure** For each problem, we allow a fixed budget of $50 \times (D + 2)$ likelihood evaluations, where $D$ is the number of variables. Given the limited number of samples, we judge the quality of the posterior approximation in terms of its first two moments, by computing the "Gaussianized" symmetrized KL divergence (gsKL) between posterior approximation and ground truth. The gsKL is defined as the symmetrized KL between two multivariate normal distributions with mean and covariances equal, respectively, to the moments of the approximate posterior and the moments of the true posterior. We measure the quality of the approximation of the LML in terms of *absolute* error from ground truth, the rationale being that differences of LML are used for model comparison. Ideally, we want the LML error to be of order 1 of less, since much larger errors could severely affect the results of a comparison (e.g., differences of LML of 10 points or more are often presented as *decisive* evidence in favor of one model [26]). On the other hand, errors $\lesssim 0.1$ can be considered negligible; higher precision is unnecessary. For each algorithm, we ran at least 20 separate runs per test problem with different random seeds, and report the median gsKL and LML error and the 95% CI of the median calculated by bootstrap. For each run, we draw the starting point $x_0$ (if requested by the algorithm) uniformly from a box within 1 prior standard deviation (SD) from the prior mean. We use the same box to define the plausible bounds for VBMC.

## 4.1 Synthetic likelihoods

**Problem set**  We built a benchmark set of synthetic likelihoods belonging to three families that represent typical features of target densities (see Supplementary Material for details). Likelihoods in the *lumpy* family are built out of a mixture of 12 multivariate normals with component means drawn randomly in the unit $D$-hypercube, distinct diagonal covariances with SDs in the $[0.2, 0.6]$ range, and mixture weights drawn from a Dirichlet distribution with unit concentration parameter. The lumpy distributions are mildly multimodal, in that modes are nearby and connected by regions with non-neglibile probability mass. In the *Student* family, the likelihood is a multivariate Student's $t$-distribution with diagonal covariance and degrees of freedom equally spaced in the $[2.5, 2 + D/2]$ range across different coordinate dimensions. These distributions have heavy tails which might be problematic for some methods. Finally, in the *cigar* family the likelihood is a multivariate normal in which one axis is 100 times longer than the others, and the covariance matrix is non-diagonal after a random rotation. The cigar family tests the ability of an algorithm to explore non axis-aligned directions. For each family, we generated test functions for $D \in \{2, 4, 6, 8, 10\}$, for a total of 15 synthetic problems. For each problem, we pick as a broad prior a multivariate normal with mean centered at the expected mean of the family of distributions, and diagonal covariance matrix with SD equal to 3-4 times the SD in each dimension. For all problems, we compute ground truth values for the LML and the posterior mean and covariance analytically or via multiple 1-D numerical integrals.

**Results**  We show the results for $D \in \{2, 6, 10\}$ in Fig. 2 (see Supplementary Material for full results, in higher resolution). Almost all algorithms perform reasonably well in very low dimension ($D = 2$), and in fact several algorithms converge faster than VBMC to the ground truth (e.g., WSABI-L). However, as we increase in dimension, we see that all algorithms start failing, with only VBMC peforming consistently well across problems. In particular, besides the simple $D = 2$ case, only VBMC obtains acceptable results for the LML with non-axis aligned distributions (*cigar*). Some algorithms (such as AGP and BAPE) exhibited large numerical instabilities on the *cigar* family, despite our best attempts at regularization, such that many runs were unable to complete.

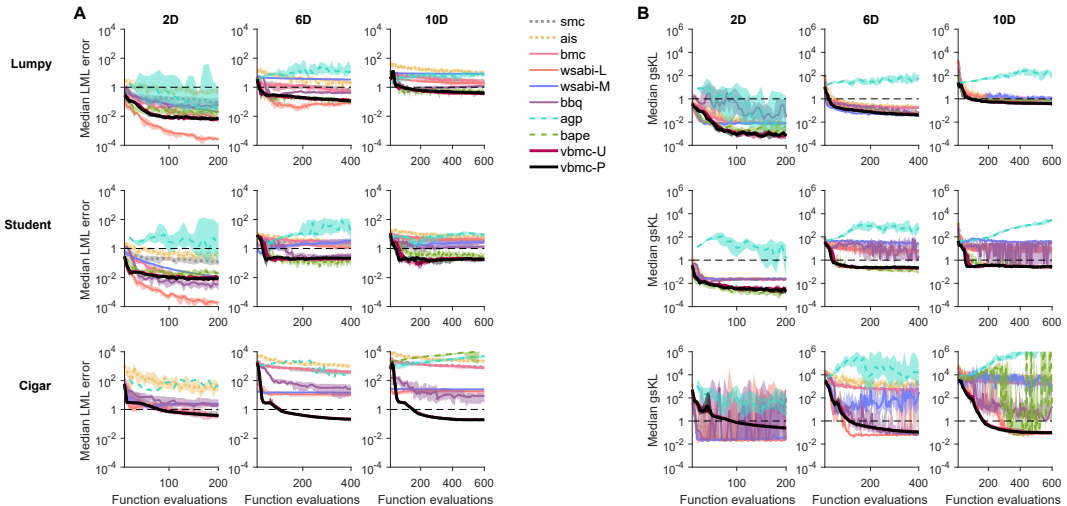

Figure 2: **Synthetic likelihoods. A.** Median absolute error of the LML estimate with respect to ground truth, as a function of number of likelihood evaluations, on the *lumpy* (top), *Student* (middle), and *cigar* (bottom) problems, for $D \in \{2, 6, 10\}$ (columns). **B.** Median "Gaussianized" symmetrized KL divergence between the algorithm's posterior and ground truth. For both metrics, shaded areas are 95 % CI of the median, and we consider a desirable threshold to be below one (dashed line).

## 4.2 Real likelihoods of neuronal model

**Problem set**  For a test with real models and data, we consider a computational model of neuronal orientation selectivity in visual cortex [14]. We fit the neural recordings of one V1 and one V2 cell with the authors' neuronal model that combines effects of filtering, suppression, and response nonlinearity [14]. The model is analytical but still computationally expensive due to large datasets and a cascade of several nonlinear operations. For the purpose of our benchmark, we fix some parameters of the original model to their MAP values, yielding an inference problem with $D = 7$ free

parameters of experimental interest. We transform bounded parameters to uncontrained space via a logit transform [24], and we place a broad Gaussian prior on each of the transformed variables, based on estimates from other neurons in the same study [14] (see Supplementary Material for more details on the setup). For both datasets, we computed the ground truth with $4 \cdot 10^5$ samples from the posterior, obtained via parallel slice sampling after a long burn-in. We calculated the ground truth LML from posterior MCMC samples via Geyer's reverse logistic regression [27], and we independently validated it with a Laplace approximation, obtained via numerical calculation of the Hessian at the MAP (for both datasets, Geyer's and Laplace's estimates of the LML are within $\sim 1$ point).

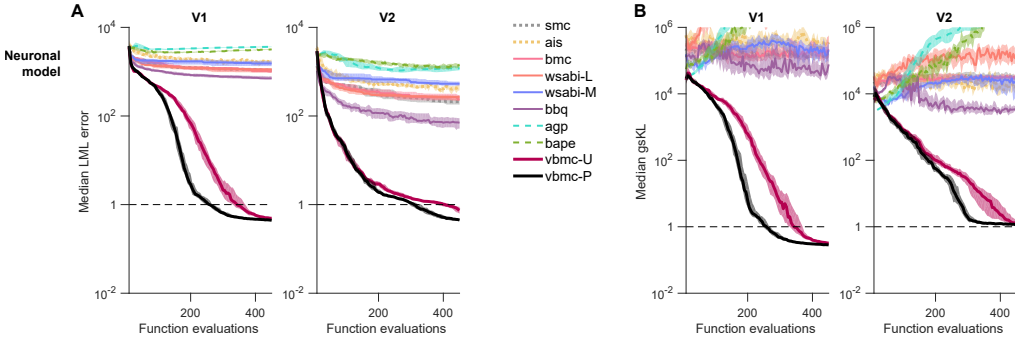

Figure 3: **Neuronal model likelihoods. A.** Median absolute error of the LML estimate, as a function of number of likelihood evaluations, for two distinct neurons ($D = 7$). **B.** Median "Gaussianized" symmetrized KL divergence between the algorithm's posterior and ground truth. See also Fig. 2.

**Results** For both datasets, VBMC is able to find a reasonable approximation of the LML and of the posterior, whereas no other algorithm produces a usable solution (Fig. 3). Importantly, the behavior of VBMC is fairly consistent across runs (see Supplementary Material). We argue that the superior results of VBMC stem from a better exploration of the posterior landscape, and from a better approximation of the log joint (used in the ELBO), related but distinct features. To show this, we first trained GPs (as we did for the other methods) on the samples collected by VBMC (see Supplementary Material). The posteriors obtained by sampling from the GPs trained on the VBMC samples scored a better gsKL than the other methods (and occasionally better than VBMC itself). Second, we estimated the marginal likelihood with WSABI-L using the samples collected by VBMC. The LML error in this hybrid approach is much lower than the error of WSABI-L alone, but still higher than the LML error of VBMC. These results combined suggest that VBMC builds better (and more stable) surrogate models and obtains higher-quality samples than the compared methods.

The performance of VBMC-U and VBMC-P is similar on synthetic functions, but the 'prospective' acquisition function converges faster on the real problem set, so we recommend $a_{\mathrm{pro}}$ as the default. Besides scoring well on quantitative metrics, VBMC is able to capture nontrivial features of the true posteriors (see Supplementary Material for examples). Moreover, VBMC achieves these results with a relatively small computational cost (see Supplementary Material for discussion).

## 5 Conclusions

In this paper, we have introduced VBMC, a novel Bayesian inference framework that combines variational inference with active-sampling Bayesian quadrature for models with expensive black-box likelihoods. Our method affords both posterior estimation and model inference by providing an approximate posterior and a lower bound to the model evidence. We have shown on both synthetic and real model-fitting problems that, given a contained budget of likelihood evaluations, VBMC is able to reliably compute valid, usable approximations in realistic scenarios, unlike previous methods whose applicability seems to be limited to very low dimension or simple likelihoods. Our method, thus, represents a novel useful tool for approximate inference in science and engineering.

We believe this is only the starting point to harness the combined power of variational inference and Bayesian quadrature. Not unlike the related field of Bayesian optimization, VBMC paves the way to a plenitude of both theoretical (e.g., analysis of convergence, development of principled acquisition functions) and applied work (e.g., application to case studies of interest, extension to noisy likelihood evaluations, algorithmic improvements), which we plan to pursue as future directions.

## Acknowledgments

We thank Robbe Goris for sharing data and code for the neuronal model; Michael Schartner and Rex Liu for comments on an earlier version of the paper; and three anonymous reviewers for useful feedback.

## Footnotes

[1]Code available at `https://github.com/lacerbi/vbmc`.

[2]We also tried a deterministic approximation of the entropy proposed in [20], with mixed results.

[3]The available code for VBMC currently supports both unbounded variables and bound constraints.

[4]We also tested BBQ* (approximate GP hyperparameter marginalization), which perfomed similarly to BBQ.

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
