[Supplementary Material · vbmc_supplementarymaterial.pdf]

# Variational Bayesian Monte Carlo — Supplementary Material

**Luigi Acerbi**[*]
Department of Basic Neuroscience
University of Geneva
`luigi.acerbi@unige.ch`

In this Supplement we include a number of derivations, implementation details, and additional results omitted from the main text.

Code used to generate the results in the paper is available at `https://github.com/lacerbi/infbench`. The VBMC algorithm is available at `https://github.com/lacerbi/vbmc`.

## Contents

---

[*]Website: `luigiacerbi.com`. Alternative e-mail: `luigi.acerbi@gmail.com`.

# A  Computing and optimizing the ELBO

For ease of reference, we recall the expression for the ELBO, for $\boldsymbol{x} \in \mathbb{R}^D$,

$$\mathcal{F}\left[q_\phi\right] = \mathbb{E}_\phi \left[\log \frac{p(\mathcal{D}|\boldsymbol{x})p(\boldsymbol{x})}{q_\phi(\boldsymbol{x})}\right] = \mathbb{E}_\phi\left[f(\boldsymbol{x})\right] + \mathcal{H}[q_\phi(\boldsymbol{x})], \tag{S1}$$

with $\mathbb{E}_\phi \equiv \mathbb{E}_{q_\phi}$, and of the variational posterior,

$$q(\boldsymbol{x}) \equiv q_\phi(\boldsymbol{x}) = \sum_{k=1}^K w_k \mathcal{N}\left(\boldsymbol{x}; \boldsymbol{\mu}_k, \sigma_k^2 \boldsymbol{\Sigma}\right), \tag{S2}$$

where $w_k$, $\boldsymbol{\mu}_k$, and $\sigma_k$ are, respectively, the mixture weight, mean, and scale of the $k$-th component, and $\boldsymbol{\Sigma} \equiv \text{diag}[\lambda^{(1)^2}, \ldots, \lambda^{(D)^2}]$ is a diagonal covariance matrix common to all elements of the mixture. The variational posterior for a given number of mixture components $K$ is parameterized by $\boldsymbol{\phi} \equiv (w_1, \ldots, w_K, \boldsymbol{\mu}_1, \ldots, \boldsymbol{\mu}_K, \sigma_1, \ldots, \sigma_K, \boldsymbol{\lambda})$.

In the following paragraphs we derive expressions for the ELBO and for its gradient. Then, we explain how we optimize it with respect to the variational parameters.

## A.1  Stochastic approximation of the entropy

We approximate the entropy of the variational distribution via simple Monte Carlo sampling as follows. Let $\mathbf{R} = \text{diag}\left[\boldsymbol{\lambda}\right]$ and $N_s$ be the number of samples per mixture component. We have

$$\begin{aligned}
\mathcal{H}\left[q(\boldsymbol{x})\right] &= -\int q(\boldsymbol{x}) \log q(\boldsymbol{x}) d\boldsymbol{x} \\
&\approx -\frac{1}{N_s} \sum_{s=1}^{N_s} \sum_{k=1}^K w_k \log q(\sigma_k \mathbf{R} \boldsymbol{\varepsilon}_{s,k} + \boldsymbol{\mu}_k) \quad \text{with} \quad \boldsymbol{\varepsilon}_{s,k} \sim \mathcal{N}\left(\mathbf{0}, \mathbb{I}_D\right) \\
&= -\frac{1}{N_s} \sum_{s=1}^{N_s} \sum_{k=1}^K w_k \log q(\boldsymbol{\xi}_{s,k}) \quad \text{with} \quad \boldsymbol{\xi}_{s,k} \equiv \sigma_k \mathbf{R} \boldsymbol{\varepsilon}_{s,k} + \boldsymbol{\mu}_k
\end{aligned} \tag{S3}$$

where we used the reparameterization trick separately for each component [1, 2]. For VBMC, we set $N_s = 100$ during the variational optimization, and $N_s = 2^{15}$ for evaluating the ELBO with high precision at the end of each iteration.

### A.1.1 Gradient of the entropy

The derivative of the entropy with respect to a variational parameter $\phi \in \{\mu, \sigma, \lambda\}$ (that is, not a mixture weight) is

$$
\begin{aligned}
\frac{d}{d\phi}\mathcal{H}\left[q(\boldsymbol{x})\right] &\approx -\frac{1}{N_{\mathrm{s}}}\sum_{s=1}^{N_{\mathrm{s}}}\sum_{k=1}^{K}w_k\frac{d}{d\phi}\log q(\boldsymbol{\xi}_{s,k})\\
&= -\frac{1}{N_{\mathrm{s}}}\sum_{s=1}^{N_{\mathrm{s}}}\sum_{k=1}^{K}w_k\left(\frac{\partial}{\partial\phi}+\sum_{i=1}^{D}\frac{d\xi_{s,k}^{(i)}}{d\phi}\frac{\partial}{\partial\xi_{s,k}^{(i)}}\right)\log q\left(\boldsymbol{\xi}_{s,k}\right)\\
&= -\frac{1}{N_{\mathrm{s}}}\sum_{s=1}^{N_{\mathrm{s}}}\sum_{k=1}^{K}\frac{w_k}{q(\boldsymbol{\xi}_{s,k})}\sum_{i=1}^{D}\frac{d\xi_{s,k}^{(i)}}{d\phi}\frac{\partial}{\partial\xi_{s,k}^{(i)}}\sum_{l=1}^{K}w_l\mathcal{N}\left(\boldsymbol{\xi}_{s,k};\boldsymbol{\mu}_l,\sigma_l^2\boldsymbol{\Sigma}\right)\\
&= \frac{1}{N_{\mathrm{s}}}\sum_{s=1}^{N_{\mathrm{s}}}\sum_{k=1}^{K}\frac{w_k}{q(\boldsymbol{\xi}_{s,k})}\sum_{i=1}^{D}\frac{d\xi_{s,k}^{(i)}}{d\phi}\sum_{l=1}^{K}w_l\frac{\xi_{s,k}^{(i)}-\mu_l^{(i)}}{\left(\sigma_k\lambda^{(i)}\right)^2}\mathcal{N}\left(\boldsymbol{\xi}_{s,k};\boldsymbol{\mu}_l,\sigma_l^2\boldsymbol{\Sigma}\right)
\end{aligned}
\tag{S4}
$$

where from the second to the third row we used the fact that the expected value of the score is zero, $\mathbb{E}_{q(\boldsymbol{\xi})}\left[\frac{\partial}{\partial\phi}\log q(\boldsymbol{\xi})\right]=0$.

In particular, for $\phi=\mu_j^{(m)}$, with $1\le m\le D$ and $1\le j\le K$,

$$
\begin{aligned}
\frac{d}{d\mu_j^{(m)}}\mathcal{H}\left[q(\boldsymbol{x})\right] &\approx -\frac{1}{N_{\mathrm{s}}}\sum_{s=1}^{N_{\mathrm{s}}}\sum_{k=1}^{K}\frac{w_k}{q(\boldsymbol{\xi}_{s,k})}\sum_{i=1}^{D}\frac{d\xi_{s,k}^{(i)}}{d\mu_j^{(m)}}\frac{\partial}{\partial\xi_{s,k}^{(i)}}\sum_{l=1}^{K}w_l\mathcal{N}\left(\boldsymbol{\xi}_{s,k};\boldsymbol{\mu}_l,\sigma_l^2\boldsymbol{\Sigma}\right)\\
&= \frac{w_j}{N_{\mathrm{s}}}\sum_{s=1}^{N_{\mathrm{s}}}\frac{1}{q(\boldsymbol{\xi}_{s,j})}\sum_{l=1}^{K}w_l\frac{\xi_{s,j}^{(m)}-\mu_l^{(m)}}{\left(\sigma_l\lambda^{(m)}\right)^2}\mathcal{N}\left(\boldsymbol{\xi}_{s,j};\boldsymbol{\mu}_l,\sigma_l^2\boldsymbol{\Sigma}\right)
\end{aligned}
\tag{S5}
$$

where we used that fact that $\frac{d\xi_{s,k}^{(i)}}{d\mu_j^{(m)}}=\delta_{im}\delta_{jk}$.

For $\phi=\sigma_j$, with $1\le j\le K$,

$$
\begin{aligned}
\frac{d}{d\sigma_j}\mathcal{H}\left[q(\boldsymbol{x})\right] &\approx -\frac{1}{N_{\mathrm{s}}}\sum_{s=1}^{N_{\mathrm{s}}}\sum_{k=1}^{K}\frac{w_k}{q(\boldsymbol{\xi}_{s,k})}\sum_{i=1}^{D}\frac{d\xi_{s,k}^{(i)}}{d\sigma_j}\frac{\partial}{\partial\xi_{s,k}^{(i)}}\sum_{l=1}^{K}w_l\mathcal{N}\left(\boldsymbol{\xi}_{s,k};\boldsymbol{\mu}_l,\sigma_l^2\boldsymbol{\Sigma}\right)\\
&= \frac{w_j}{K^2N_{\mathrm{s}}}\sum_{s=1}^{N_{\mathrm{s}}}\frac{1}{q(\boldsymbol{\xi}_{s,j})}\sum_{i=1}^{D}\lambda^{(i)}\varepsilon_{s,j}^{(i)}\sum_{l=1}^{K}w_l\frac{\xi_{s,j}^{(i)}-\mu_l^{(i)}}{\left(\sigma_l\lambda^{(i)}\right)^2}\mathcal{N}\left(\boldsymbol{\xi}_{s,j};\boldsymbol{\mu}_l,\sigma_l^2\boldsymbol{\Sigma}\right)
\end{aligned}
\tag{S6}
$$

where we used that fact that $\frac{d\xi_{s,k}^{(i)}}{d\sigma_j}=\lambda^{(i)}\varepsilon_{s,j}^{(i)}\delta_{jk}$.

For $\phi=\lambda^{(m)}$, with $1\le m\le D$,

$$
\begin{aligned}
\frac{d}{d\lambda^{(m)}}\mathcal{H}\left[q(\boldsymbol{x})\right] &\approx -\frac{1}{N_{\mathrm{s}}}\sum_{s=1}^{N_{\mathrm{s}}}\sum_{k=1}^{K}\frac{w_k}{q(\boldsymbol{\xi}_{s,k})}\sum_{i=1}^{D}\frac{d\xi_{s,k}^{(i)}}{d\lambda^{(m)}}\frac{\partial}{\partial\xi_{s,k}^{(i)}}\sum_{l=1}^{K}w_l\mathcal{N}\left(\boldsymbol{\xi}_{s,k};\boldsymbol{\mu}_l,\sigma_l^2\boldsymbol{\Sigma}\right)\\
&= \frac{1}{N_{\mathrm{s}}}\sum_{s=1}^{N_{\mathrm{s}}}\sum_{k=1}^{K}\frac{w_k\sigma_k\varepsilon_{s,k}^{(m)}}{q(\boldsymbol{\xi}_{s,k})}\sum_{l=1}^{K}w_l\frac{\xi_{s,k}^{(m)}-\mu_l^{(m)}}{\left(\sigma_l\lambda^{(m)}\right)^2}\mathcal{N}\left(\boldsymbol{\xi}_{s,k};\boldsymbol{\mu}_l,\sigma_l^2\boldsymbol{\Sigma}\right)
\end{aligned}
\tag{S7}
$$

where we used that fact that $\frac{d\xi_{s,k}^{(i)}}{d\lambda^{(m)}}=\sigma_k\varepsilon_{s,k}^{(i)}\delta_{im}$.

Finally, the derivative with respect to variational mixture weight $w_j$, for $1\le j\le K$, is

$$
\frac{\partial}{\partial w_j}\mathcal{H}\left[q(\boldsymbol{x})\right]\approx -\frac{1}{N_{\mathrm{s}}}\sum_{s=1}^{N_{\mathrm{s}}}\left[\log q(\boldsymbol{\xi}_{s,j})+\sum_{k=1}^{K}\frac{w_k}{q(\boldsymbol{\xi}_{s,k})}q_j(\boldsymbol{\xi}_{s,k})\right].
\tag{S8}
$$

## A.2 Expected log joint

For the expected log joint we have

$$
\begin{aligned}
\mathcal{G}[q(\boldsymbol{x})] = \mathbb{E}_{\boldsymbol{\phi}}\left[f(\boldsymbol{x})\right] &= \sum_{k=1}^{K} w_k \int \mathcal{N}\left(\boldsymbol{x}; \boldsymbol{\mu}_k, \sigma_k^2 \boldsymbol{\Sigma}\right) f(\boldsymbol{x}) d\boldsymbol{x} \\
&= \sum_{k=1}^{K} w_k \mathcal{I}_k.
\end{aligned}
\tag{S9}
$$

To solve the integrals in Eq. S9 we approximate $f(\boldsymbol{x})$ with a Gaussian process (GP) with a *squared exponential* (that is, rescaled Gaussian) covariance function,

$$
\mathbf{K}_{pq} = \kappa\left(\boldsymbol{x}_p, \boldsymbol{x}_q\right) = \sigma_f^2 \Lambda \mathcal{N}\left(\boldsymbol{x}_p; \boldsymbol{x}_q, \boldsymbol{\Sigma}_\ell\right) \qquad \text{with } \boldsymbol{\Sigma}_\ell = \text{diag}\left[\ell^{(1)^2}, \ldots, \ell^{(D)^2}\right],
\tag{S10}
$$

where $\Lambda \equiv (2\pi)^{\frac{D}{2}} \prod_{i=1}^{D} \ell^{(i)}$ is equal to the normalization factor of the Gaussian.[1] For the GP we also assume a Gaussian likelihood with observation noise variance $\sigma_{\text{obs}}^2$ and, for the sake of exposition, a *constant* mean function $m \in \mathbb{R}$. We will later consider the case of a *negative quadratic* mean function, as per the main text.

### A.2.1 Posterior mean of the integral and its gradient

The posterior predictive mean of the GP, given training data $\boldsymbol{\Xi} = \{\mathbf{X}, \boldsymbol{y}\}$, where $\mathbf{X}$ are $n$ training inputs with associated observed values $\boldsymbol{y}$, is

$$
\overline{f}(\boldsymbol{x}) = \kappa(\boldsymbol{x}, \mathbf{X}) \left[\kappa(\mathbf{X}, \mathbf{X}) + \sigma_{\text{obs}}^2 \mathbf{I}_n\right]^{-1} (\boldsymbol{y} - m) + m.
\tag{S11}
$$

Thus, for each integral in Eq. S9 we have in expectation over the GP posterior

$$
\begin{aligned}
\mathbb{E}_{f|\boldsymbol{\Xi}}\left[\mathcal{I}_k\right] &= \int \mathcal{N}\left(\boldsymbol{x}; \boldsymbol{\mu}_k, \sigma_k^2 \boldsymbol{\Sigma}\right) \overline{f}(\boldsymbol{x}) d\boldsymbol{x} \\
&= \left[\sigma_f^2 \int \mathcal{N}\left(\boldsymbol{x}; \boldsymbol{\mu}_k, \sigma_k^2 \boldsymbol{\Sigma}\right) \mathcal{N}\left(\boldsymbol{x}; \mathbf{X}, \boldsymbol{\Sigma}_\ell\right) d\boldsymbol{x}\right] \left[\kappa(\mathbf{X}, \mathbf{X}) + \sigma_{\text{obs}}^2 \mathbf{I}\right]^{-1} (\boldsymbol{y} - m) + m \\
&= \boldsymbol{z}_k^\top \left[\kappa(\mathbf{X}, \mathbf{X}) + \sigma_{\text{obs}}^2 \mathbf{I}\right]^{-1} (\boldsymbol{y} - m) + m,
\end{aligned}
\tag{S12}
$$

where $\boldsymbol{z}_k$ is a $n$-dimensional vector with entries $z_k^{(p)} = \sigma_f^2 \mathcal{N}\left(\boldsymbol{\mu}_k; \boldsymbol{x}_p, \sigma_k^2 \boldsymbol{\Sigma} + \boldsymbol{\Sigma}_\ell\right)$ for $1 \le p \le n$. In particular, defining $\tau_k^{(i)} \equiv \sqrt{\sigma_k^2 \lambda^{(i)^2} + \ell^{(i)^2}}$ for $1 \le i \le D$,

$$
z_k^{(p)} = \frac{\sigma_f^2}{(2\pi)^{\frac{D}{2}} \prod_{i=1}^{D} \tau_k^{(i)}} \exp\left\{-\frac{1}{2} \sum_{i=1}^{D} \frac{\left(\mu_k^{(i)} - \boldsymbol{x}_p^{(i)}\right)^2}{\tau_k^{(i)^2}}\right\}.
\tag{S13}
$$

We can compute derivatives with respect to the variational parameters $\phi \in (\mu, \sigma, \lambda)$ as

$$
\begin{aligned}
\frac{\partial}{\partial \mu_j^{(l)}} z_k^{(p)} &= \delta_{jk} \frac{\boldsymbol{x}_p^{(l)} - \mu_k^{(l)}}{\tau_k^{(l)^2}} z_k^{(p)} \\
\frac{\partial}{\partial \sigma_j} z_k^{(p)} &= \delta_{jk} \sum_{i=1}^{D} \frac{\lambda^{(i)^2}}{\tau_k^{(i)^2}} \left[\frac{\left(\mu_k^{(i)} - \boldsymbol{x}_p^{(i)}\right)^2}{\tau_k^{(i)^2}} - 1\right] \sigma_k z_k^{(p)} \\
\frac{\partial}{\partial \lambda^{(l)}} z_k^{(p)} &= \frac{\sigma_k^2}{\tau_k^{(l)^2}} \left[\frac{\left(\mu_k^{(l)} - \boldsymbol{x}_p^{(l)}\right)^2}{\tau_k^{(l)^2}} - 1\right] \lambda^{(l)} z_k^{(p)}
\end{aligned}
\tag{S14}
$$

The derivative of Eq. S9 with respect to mixture weight $w_k$ is simply $\mathcal{I}_k$.

### A.2.2 Posterior variance of the integral

We compute the variance of Eq. S9 under the GP approximation as [3]

$$
\begin{aligned}
\mathrm{Var}_{f|\mathcal{X}}[\mathcal{G}] &= \int\int q(\boldsymbol{x})q(\boldsymbol{x}')C_{\boldsymbol{\Xi}}\left(f(\boldsymbol{x}),f(\boldsymbol{x}')\right)\,d\boldsymbol{x}d\boldsymbol{x}' \\
&= \sum_{j=1}^{K}\sum_{k=1}^{K}w_jw_k\int\int\mathcal{N}\left(\boldsymbol{x};\boldsymbol{\mu}_j,\sigma_j^2\boldsymbol{\Sigma}\right)\mathcal{N}\left(\boldsymbol{x}';\boldsymbol{\mu}_k,\sigma_k^2\boldsymbol{\Sigma}\right)C_{\boldsymbol{\Xi}}\left(f(\boldsymbol{x}),f(\boldsymbol{x}')\right)\,d\boldsymbol{x}d\boldsymbol{x}' \\
&= \sum_{j=1}^{K}\sum_{k=1}^{K}w_jw_k\mathcal{J}_{jk}
\end{aligned}
\tag{S15}
$$

where $C_{\boldsymbol{\Xi}}$ is the GP posterior predictive covariance,

$$
C_{\boldsymbol{\Xi}}\left(f(\boldsymbol{x}),f(\boldsymbol{x}')\right) = \kappa(\boldsymbol{x},\boldsymbol{x}') - \kappa(\boldsymbol{x},\mathbf{X})\left[\kappa(\mathbf{X},\mathbf{X})+\sigma_{\mathrm{obs}}^2\mathbf{I}_n\right]^{-1}\kappa(\mathbf{X},\boldsymbol{x}').
\tag{S16}
$$

Thus, each term in Eq. S15 can be written as

$$
\begin{aligned}
\mathcal{J}_{jk} &= \int\int\mathcal{N}\left(\boldsymbol{x};\boldsymbol{\mu}_j,\sigma_j^2\boldsymbol{\Sigma}\right)\left[\sigma_f^2\mathcal{N}\left(\boldsymbol{x};\boldsymbol{x}',\boldsymbol{\Sigma}_\ell\right)-\sigma_f^2\mathcal{N}\left(\boldsymbol{x};\mathbf{X},\boldsymbol{\Sigma}_\ell\right)\left[\kappa(\mathbf{X},\mathbf{X})+\sigma_{\mathrm{obs}}^2\mathbf{I}_n\right]^{-1}\sigma_f^2\mathcal{N}\left(\mathbf{X};\boldsymbol{x}',\boldsymbol{\Sigma}_\ell\right)\right]\times \\
&\quad \times\mathcal{N}\left(\boldsymbol{x}';\boldsymbol{\mu}_k,\sigma_k^2\boldsymbol{\Sigma}\right)\,d\boldsymbol{x}d\boldsymbol{x}' \\
&= \sigma_f^2\mathcal{N}\left(\boldsymbol{\mu}_j;\boldsymbol{\mu}_k,\boldsymbol{\Sigma}_\ell+(\sigma_j^2+\sigma_k^2)\boldsymbol{\Sigma}\right)-\boldsymbol{z}_j^\top\left[\kappa(\mathbf{X},\mathbf{X})+\sigma_{\mathrm{obs}}^2\mathbf{I}_n\right]^{-1}\boldsymbol{z}_k.
\end{aligned}
\tag{S17}
$$

### A.2.3 Negative quadratic mean function

We consider now a GP with a *negative quadratic* mean function,

$$
m(\boldsymbol{x}) \equiv m_{\mathrm{NQ}}(\boldsymbol{x}) = m_0 - \frac{1}{2}\sum_{i=1}^{D}\frac{\left(x^{(i)}-x_{\mathrm{m}}^{(i)}\right)^2}{\omega^{(i)2}}.
\tag{S18}
$$

With this mean function, for each integral in Eq. S9 we have in expectation over the GP posterior,

$$
\begin{aligned}
\mathbb{E}_{f|\boldsymbol{\Xi}}[\mathcal{I}_k] &= \int\mathcal{N}\left(\boldsymbol{x};\boldsymbol{\mu}_k,\sigma_k^2\boldsymbol{\Sigma}\right)\left[\sigma_f^2\mathcal{N}\left(\boldsymbol{x};\mathbf{X},\boldsymbol{\Sigma}_\ell\right)\left[\kappa(\mathbf{X},\mathbf{X})+\sigma_{\mathrm{obs}}^2\mathbf{I}\right]^{-1}(\boldsymbol{y}-m(\mathbf{X}))+m(\boldsymbol{x})\right]d\boldsymbol{x} \\
&= \boldsymbol{z}_k^\top\left[\kappa(\mathbf{X},\mathbf{X})+\sigma_{\mathrm{obs}}^2\mathbf{I}\right]^{-1}(\boldsymbol{y}-m(\mathbf{X}))+m_0+\nu_k,
\end{aligned}
\tag{S19}
$$

where we defined

$$
\nu_k = -\frac{1}{2}\sum_{i=1}^{D}\frac{1}{\omega^{(i)2}}\left(\mu_k^{(i)2}+\sigma_k^2\lambda^{(i)2}-2\mu_k^{(i)}x_{\mathrm{m}}^{(i)}+x_{\mathrm{m}}^{(i)2}\right).
\tag{S20}
$$

## A.3 Optimization of the approximate ELBO

In the following paragraphs we describe how we optimize the ELBO in each iteration of VBMC, so as to find the variational posterior that best approximates the current GP model of the posterior.

### A.3.1 Reparameterization

For the purpose of the optimization, we reparameterize the variational parameters such that they are defined in a potentially unbounded space. The mixture means, $\boldsymbol{\mu}_k$, remain the same. We switch from mixture scale parameters $\sigma_k$ to their logarithms, $\log\sigma_k$, and similarly from coordinate length scales, $\lambda^{(i)}$, to $\log\lambda^{(i)}$. Finally, we parameterize mixture weights as unbounded variables, $\eta_k\in\mathbb{R}$, such that $w_k\equiv e^{\eta_k}/\sum_l e^{\eta_l}$ (softmax function). We compute the appropriate Jacobian for the change of variables and apply it to the gradients calculated in Sections A.1 and A.2.

### A.3.2 Choice of starting points

In each iteration, we first perform a quick exploration of the ELBO landscape in the vicinity of the current variational posterior by generating $n_{\text{fast}} \cdot K$ candidate starting points, obtained by randomly jittering, rescaling, and reweighting components of the current variational posterior. In this phase we also add new mixture components, if so requested by the algorithm, by randomly splitting and jittering existing components. We evaluate the ELBO at each candidate starting point, and pick the point with the best ELBO as starting point for the subsequent optimization.

For most iterations we use $n_{\text{fast}} = 5$, except for the first iteration and the first iteration after the end of warm-up, for which we set $n_{\text{fast}} = 50$.

### A.3.3 Stochastic gradient descent

We optimize the (negative) ELBO via stochastic gradient descent, using a customized version of Adam [4]. Our modified version of Adam includes a time-decaying learning rate, defined as

$$\alpha_t = \alpha_{\text{min}} + (\alpha_{\text{max}} - \alpha_{\text{min}}) \exp\left[-\frac{t}{\tau}\right] \tag{S21}$$

where $t$ is the current iteration of the optimizer, $\alpha_{\text{min}}$ and $\alpha_{\text{max}}$ are, respectively, the minimum and maximum learning rate, and $\tau$ is the decay constant. We stop the optimization when the estimated change in function value or in the parameter vector across the past $n_{\text{batch}}$ iterations of the optimization goes below a given threshold.

We set as hyperparameters of the optimizer $\beta_1 = 0.9$, $\beta_2 = 0.99$, $\epsilon \approx 1.49 \cdot 10^{-8}$ (square root of double precision), $\alpha_{\text{min}} = 0.001$, $\tau = 200$, $n_{\text{batch}} = 20$. We set $\alpha_{\text{max}} = 0.1$ during warm-up, and $\alpha_{\text{max}} = 0.01$ thereafter.

## B    Algorithmic details

We report here several implementation details of the VBMC algorithm omitted from the main text.

### B.1    Regularization of acquisition functions

Active sampling in VBMC is performed by maximizing an acquisition function $a : \mathcal{X} \subseteq \mathbb{R}^D \to [0, \infty)$, where $\mathcal{X}$ is the support of the target density. In the main text we describe two such functions, uncertainty sampling ($a_{\text{us}}$) and prospective uncertainty sampling ($a_{\text{pro}}$).

A well-known problem with GPs, in particular when using smooth kernels such as the squared exponential, is that they become numerically unstable when the training set contains points which are too close to each other, producing a ill-conditioned Gram matrix. Here we reduce the chance of this happening by introducing a correction factor as follows. For any acquisition function $a$, its regularized version $a^{\text{reg}}$ is defined as

$$a^{\text{reg}}(\boldsymbol{x}) = a(\boldsymbol{x}) \exp\left\{-\left(\frac{V^{\text{reg}}}{V_{\boldsymbol{\Xi}}(\boldsymbol{x})} - 1\right) |[V_{\boldsymbol{\Xi}}(\boldsymbol{x}) < V^{\text{reg}}]|\right\} \tag{S22}$$

where $V_{\boldsymbol{\Xi}}(\boldsymbol{x})$ is the total posterior predictive variance of the GP at $\boldsymbol{x}$ for the given training set $\boldsymbol{\Xi}$, $V^{\text{reg}}$ a regularization parameter, and we denote with $|[\cdot]|$ *Iverson's bracket* [5], which takes value 1 if the expression inside the bracket is true, 0 otherwise. Eq. S22 enforces that the regularized acquisition function does not pick points too close to points in $\boldsymbol{\Xi}$. For VBMC, we set $V^{\text{reg}} = 10^{-4}$.

## B.2 GP hyperparameters and priors

The GP model in VBMC has $3D + 3$ hyperparameters, $\boldsymbol{\psi} = (\boldsymbol{\ell}, \sigma_f, \sigma_{\text{obs}}, m_0, \boldsymbol{x}_{\text{m}}, \boldsymbol{\omega})$. We define all scale hyperparameters, that is $\{\boldsymbol{\ell}, \sigma_f, \sigma_{\text{obs}}, \boldsymbol{\omega}\}$, in log space.

We assume independent priors on each hyperparameter. For some hyperparameters, we impose as prior a broad Student's $t$ distribution with a given mean $\mu$, scale $\sigma$, and $\nu = 3$ degrees of freedom. Following an empirical Bayes approach, mean and scale of the prior might depend on the current training set. For all other hyperparameters we assume a uniform flat prior. GP hyperparameters and their priors are reported in Table S1.

| Hyperparameter | Description | Prior mean $\mu$ | Prior scale $\sigma$ |
|---|---|---|---|
| $\log \ell^{(i)}$ | Input length scale ($i$-th dimension) | $\log \text{SD}\left[\mathbf{X}_{\text{hpd}}^{(i)}\right]$ | $\max\left\{2, \log \frac{\text{diam}\left[\mathbf{X}_{\text{hpd}}^{(i)}\right]}{\text{SD}\left[\mathbf{X}_{\text{hpd}}^{(i)}\right]}\right\}$ |
| $\log \sigma_f$ | Output scale | Uniform | — |
| $\log \sigma_{\text{obs}}$ | Observation noise | $\log 0.001$ | $0.5$ |
| $m_0$ | Mean function maximum | $\max \boldsymbol{y}_{\text{hpd}}$ | $\text{diam}\left[\boldsymbol{y}_{\text{hpd}}\right]$ |
| $x_{\text{m}}^{(i)}$ | Mean function location ($i$-th dim.) | Uniform | — |
| $\log \omega^{(i)}$ | Mean function length scale ($i$-th dim.) | Uniform | — |

Table S1: GP hyperparameters and their priors. See text for more information.

In Table S1, $\text{SD}[\cdot]$ denotes the sample standard deviation and $\text{diam}[\cdot]$ the *diameter* of a set, that is the maximum element minus the minimum. We define the *high posterior density* training set, $\boldsymbol{\Xi}_{\text{hpd}} = \{\mathbf{X}_{\text{hpd}}, \boldsymbol{y}_{\text{hpd}}\}$, constructed by keeping a fraction $f_{\text{hpd}}$ of the training points with highest target density values. For VBMC, we use $f_{\text{hpd}} = 0.8$ (that is, we only ignore a small fraction of the points in the training set).

## B.3 Transformation of variables

In VBMC, the problem coordinates are defined in an unbounded internal working space, $\boldsymbol{x} \in \mathbb{R}^D$. All original problem coordinates $x_{\text{orig}}^{(i)}$ for $1 \leq i \leq D$ are independently transformed by a mapping $g_i : \mathcal{X}_{\text{orig}}^{(i)} \to \mathbb{R}$ defined as follows.

Unbounded coordinates are 'standardized' with respect to the plausible box, $g_{\text{unb}}(x_{\text{orig}}) = \frac{x_{\text{orig}} - (\text{PLB} + \text{PUB})/2}{\text{PUB} - \text{PLB}}$, where PLB and PUB are here, respectively, the plausible lower bound and plausible upper bound of the coordinate under consideration.

Bounded coordinates are first mapped to an unbounded space via a logit transform, $g_{\text{bnd}}(x_{\text{orig}}) = \log\left(\frac{z}{1-z}\right)$ with $z = \frac{x_{\text{orig}} - \text{LB}}{\text{UB} - \text{LB}}$, where LB and UB are here, respectively, the lower and upper bound of the coordinate under consideration. The remapped variables are then 'standardized' as above, using the remapped PLB and PUB values after the logit transform.

Note that probability densities are transformed under a change of coordinates by a multiplicative factor equal to the inverse of the determinant of the Jacobian of the transformation. Thus, the value of the observed log joint $y$ used by VBMC relates to the value $y_{\text{orig}}$ of the log joint density, observed in the original (untransformed) coordinates, as follows,

$$y(\boldsymbol{x}) = y^{\text{orig}}(\boldsymbol{x}_{\text{orig}}) - \sum_{i=1}^{D} \log g_i'(\boldsymbol{x}_{\text{orig}}), \tag{S23}$$

where $g_i'$ is the derivative of the transformation for the $i$-th coordinate, and $\boldsymbol{x} = g(\boldsymbol{x}_{\text{orig}})$. See for example [6] for more information on transformations of variables.

## B.4 Termination criteria

The VBMC algorithm terminates when reaching a maximum number of target density evaluations, or when achieving long-term stability of the variational solution, as described below.

### B.4.1 Reliability index

At the end of each iteration $t$ of the VBMC algorithm, we compute a set of reliability features of the current variational solution.

1. The absolute change in mean ELBO from the previous iteration:

$$\rho_1(t) = \frac{|\mathbb{E}\left[\text{ELBO}(t)\right] - \mathbb{E}\left[\text{ELBO}(t-1)\right]|}{\Delta_{\text{SD}}},\qquad\text{(S24)}$$

   where $\Delta_{\text{SD}} > 0$ is a tolerance parameter on the error of the ELBO.

2. The uncertainty of the current ELBO:

$$\rho_2(t) = \frac{\sqrt{\mathbb{V}\left[\text{ELBO}(t)\right]}}{\Delta_{\text{SD}}}.\qquad\text{(S25)}$$

3. The change in symmetrized KL divergence between the current variational posterior $q_t \equiv q_{\boldsymbol{\phi}_t}(\boldsymbol{x})$ and the one from the previous iteration:

$$\rho_3(t) = \frac{\text{KL}(q_t||q_{t-1}) + \text{KL}(q_{t-1}||q_t)}{2\Delta_{\text{KL}}},\qquad\text{(S26)}$$

   where for Eq. S26 we use the Gaussianized KL divergence (that is, we compare solutions only based on their mean and covariance), and $\Delta_{\text{KL}} > 0$ is a tolerance parameter for differences in variational posterior.

The parameters $\Delta_{\text{SD}}$ and $\Delta_{\text{KL}}$ are chosen such that $\rho_j \lesssim 1$, with $j = 1, 2, 3$, for features that are deemed indicative of a good solution. For VBMC, we set $\Delta_{\text{SD}} = 0.1$ and $\Delta_{\text{KL}} = 0.01 \cdot \sqrt{D}$.

The *reliability index* $\rho(t)$ at iteration $t$ is obtained by averaging the individual reliability features $\rho_j(t)$.

### B.4.2 Long-term stability termination condition

The long-term stability termination condition is reached at iteration $t$ when:

1. all reliability features $\rho_j(t)$ are below 1;
2. the reliability index $\rho$ has remained below 1 for the past $n_{\text{stable}}$ iterations (with the exception of at most one iteration, excluding the current one);
3. the slope of the ELCBO computed across the past $n_{\text{stable}}$ iterations is below a given threshold $\Delta_{\text{IMPRO}} > 0$, suggesting that the ELCBO is stationary.

For VBMC, we set by default $n_{\text{stable}} = 8$ and $\Delta_{\text{IMPRO}} = 0.01$. For computing the ELCBO we use $\beta_{\text{LCB}} = 3$ (see Eq. 8 in the main text).

### B.4.3 Validation of VBMC solutions

Long-term stability of the variational solution is *suggestive* of convergence of the algorithm to a (local) optimum, but it should not be taken as a conclusive result without further validation. In fact, without additional information, there is no way to know whether the algorithm has converged to a good solution, let alone to the global optimum. For this reason, we recommend to run the algorithm multiple times and compare the solutions, and to perform posterior predictive checks [7]. See also [8] for a discussion of methods to validate the results of variational inference.

# C   Experimental details and additional results

## C.1   Synthetic likelihoods

We plot in Fig. S1 synthetic target densities belonging to the test families described in the main text (*lumpy*, *Student*, *cigar*), for the $D = 2$ case. We also plot examples of solutions returned by VBMC after reaching long-term stability, and indicate the number of iterations.

Figure S1: **Synthetic target densities and example solutions.** *Top:* Contour plots of two-dimensional synthetic target densities. *Bottom:* Contour plots of example variational posteriors returned by VBMC, and iterations until convergence.

Note that VBMC, despite being overall the best-performing algorithm on the *cigar* family in higher dimensions, still underestimates the variance along the major axis of the distribution. This is because the variational mixture components have axis-aligned (diagonal) covariances, and thus many mixture components are needed to approximate non-axis aligned densities. Future work should investigate alternative representations of the variational posterior to increase the expressive power of VBMC, while keeping its computational efficiency and stability.

We plot in Fig. S2 the performance of selected algorithms on the synthetic test functions, for $D \in \{2, 4, 6, 8, 10\}$. These results are the same as those reported in Fig. 2 in the main text, but with higher resolution. To avoid clutter, we exclude algorithms with particularly poor performance or whose plots are redundant with others. In particular, the performance of VBMC-U is virtually identical to VBMC-P here, so we only report the latter. Analogously, with a few minor exceptions, WSABI-M performs similarly or worse than WSABI-L across all problems. AIS suffers from the lack of problem-specific tuning, performing no better than SMC here, and the AGP algorithm diverges on most problems. Finally, we did not manage to get BAPE to run on the *cigar* family, for $D \leq 6$, without systematically incurring in numerical issues with the GP approximation (with and without regularization of the BAPE acquisition function, as per Section B.1), so these plots are missing.

## C.2   Neuronal model

As a real model-fitting problem, we considered in the main text a neuronal model that combines effects of filtering, suppression, and response nonlinearity, applied to two real data sets (one V1 and one V2 neurons) [9]. The purpose of the original study was to explore the origins of diversity of neuronal orientation selectivity in visual cortex via a combination of novel stimuli (orientation mixtures) and modeling [9]. This model was also previously considered as a case study for a benchmark of Bayesian optimization and other black-box optimization algorithms [10].

Figure S2: **Full results on synthetic likelihoods. A.** Median absolute error of the LML estimate with respect to ground truth, as a function of number of likelihood evaluations, on the *lumpy* (top), *Student* (middle), and *cigar* (bottom) problems, for $D \in \{2, 4, 6, 8, 10\}$ (columns). **B.** Median "Gaussianized" symmetrized KL divergence between the algorithm's posterior and ground truth. For both metrics, shaded areas are 95 % CI of the median, and we consider a desirable threshold to be below one (dashed line). This figure reproduces Fig. 2 in the main text with more details. Note that panels here may have different vertical axes.

### C.2.1 Model parameters

In total, the original model has 12 free parameters: 5 parameters specifying properties of a linear filtering mechanism, 2 parameters specifying nonlinear transformation of the filter output, and 5 parameters controlling response range and amplitude. For the analysis in the main text, we considered a subset of $D = 7$ parameters deemed 'most interesting' by the authors of the original study [9], while fixing the others to their MAP values found by our previous optimization benchmark [10].

The seven model parameters of interest from the original model, their ranges, and the chosen plausible bounds are reported in Table S2.

| Parameter | Description | LB | UB | PLB | PUB |
|-----------|-------------|-----|-----|-----|-----|
| $x_1$ | Preferred direction of motion (deg) | 0 | 360 | 90 | 270 |
| $x_2$ | Preferred spatial frequency (cycles per deg) | 0.05 | 15 | 0.5 | 10 |
| $x_3$ | Aspect ratio of 2-D Gaussian | 0.1 | 3.5 | 0.3 | 3.2 |
| $x_4$ | Derivative order in space | 0.1 | 3.5 | 0.3 | 3.2 |
| $x_5$ | Gain inhibitory channel | -1 | 1 | -0.3 | 0.3 |
| $x_6$ | Response exponent | 1 | 6.5 | 2 | 5 |
| $x_7$ | Variance of response gain | 0.001 | 10 | 0.01 | 1 |

Table S2: Parameters and bounds of the neuronal model (before remapping).

Since all original parameters are bounded, for the purpose of our analysis we remapped them to an unbounded space via a shifted and rescaled logit transform, correcting the value of the log posterior with the log Jacobian (see Section B.3). For each parameter, we set independent Gaussian priors in the transformed space with mean equal to the average of the transformed values of PLB and PUB (see Table S2), and with standard deviation equal to half the plausible range in the transformed space.

### C.2.2 True and approximate posteriors

We plot in Fig. S3 the 'true' posterior obtained via extensive MCMC sampling for one of the two datasets (V2 neuron), and we compare it with an example variational solution returned by VBMC after reaching long-term stability (here in 52 iterations, which correspond to 260 target density evaluations).

We note that VBMC obtains a good approximation of the true posterior, which captures several features of potential interest, such as the correlation between the inhibition gain ($x_5$) and response exponent ($x_6$), and the skew in the preferred spatial frequency ($x_2$). The variational posterior, however, misses some details, such as the long tail of the aspect ratio ($x_3$), which is considerably thinner in the approximation than in the true posterior.

Figure S3: **True and approximate posterior of neuronal model (V2 neuron).** *Top*: Triangle plot of the 'true' posterior (obtained via MCMC) for the neuronal model applied to the V2 neuron dataset. Each panel below the diagonal is the contour plot of the 2-D marginal distribution for a given parameter pair. Panels on the diagonal are histograms of the 1-D marginal distribution of the posterior for each parameter. *Bottom*: Triangle plot of a typical variational solution returned by VBMC.

# D   Analysis of VBMC

In this section we report additional analyses of the VBMC algorithm.

## D.1   Variability between VBMC runs

In the main text we have shown the median performance of VBMC, but a crucial question for a practical application of the algorithm is the amount of variability between runs, due to stochasticity in the algorithm and choice of starting point (in this work, drawn uniformly randomly inside the plausible box). We plot in Fig. S4 the performance of one hundred runs of VBMC on the neuronal model datasets, together with the 50th (the median), 75th, and 90th percentiles. The performance of VBMC on this real problem is fairly robust, in that some runs take longer but the majority of them converges to quantitatively similar solutions.

Figure S4: **Variability of VBMC performance.  A.** Absolute error of the LML estimate, as a function of number of likelihood evaluations, for the two neuronal datasets. Each grey line is one of 100 distinct runs of VBMC. Thicker lines correspond to the 50th (median), 75th, and 90th percentile across runs (the median is the same as in Fig. 3 in the main text). **B.** "Gaussianized" symmetrized KL divergence between the algorithm's posterior and ground truth, for 100 distinct runs of VBMC. See also Fig. 3 in the main text.

## D.2   Computational cost

The computational cost of VBMC stems in each iteration of the algorithm primarily from three sources: active sampling, GP training, and variational optimization. Active sampling requires repeated computation of the acquisition function (for its optimization), whose cost is dominated by calculation of the posterior predictive variance of the GP, which scales as $O(n^2)$, where $n$ is the number of training points. GP training scales as $O(n^3)$, due to inversion of the Gram matrix. Finally, variational optimization scales as $O(Kn)$, where $K$ is the number of mixture components. In practice, we found in many cases that in early iterations the costs are equally divided between the three phases, but later on both GP training and variational optimization dominate the algorithmic cost. In particular, the number of components $K$ has a large impact on the effective cost.

As an example, we plot in Fig. S5 the algorithmic cost per function evaluation of different inference algorithms that have been run on the V1 neuronal dataset (algorithmic costs are similar for the V2 dataset). We consider only methods which use active sampling with a reasonable performance on at least some of the problems. We define as algorithmic cost the time spent inside the algorithm, ignoring the time used to evaluate the log likelihood function. For comparison, evaluation of the log likelihood of this problem takes about 1 s on the reference laptop computer we used. Note that for the WSABI and BBQ algoritms, the algorithmic cost reported here does not include the additional computational cost of sampling an approximate distrbution from the GP posterior (WSABI and BBQ, per se, only compute an approximation of the marginal likelihood).

VBMC on this problem exhibits a moderate cost of 2-3 s per function evaluation, when averaged across the entire run. Moreover, many runs would converge within 250-300 function evaluations, as shown in Figure S4, further lowering the effective cost per function evaluation. For the considered budget of function evaluations, WSABI (in particular, WSABI-L) is up to one order of magnitude faster than VBMC. This speed is remarkable, although it does not offset the limited performance of

Figure S5: **Algorithmic cost per function evaluation.** Median algorithmic cost per function evaluation, as a function of number of likelihood function evaluations, for different algorithms performing inference over the V1 neuronal dataset. Shaded areas are 95 % CI of the median.

the algorithm on more complex problems. WSABI-M is generally more expensive than WSABI-L (even though still quite fast), with a similar or slightly worse performance. Here our implementation of BAPE results to be slightly more expensive than VBMC. Perhaps it is possible to obtain faster implementations of BAPE, but, even so, the quality of solutions would still not match that of VBMC (also, note the general instability of the algorithm). Finally, we see that BBQ incurs in a massive algorithmic cost due to the complex GP approximation and expensive acquisition function used. Notably, the solutions obtained by BBQ in our problem sets are relatively good compared to the other algorithms, but still substantially worse than VBMC on all but the easiest problems, despite the much larger computational overhead.

The dip in cost that we observe in VBMC at around 275 function evaluations is due to the switch from GP hyperparameter sampling to optimization. The cost of BAPE oscillates because of the cost of retraining the GP model and MCMC sampling from the approximate posterior every 10 function evaluations. Similarly, by default BBQ retrains the GP model ten times, logarithmically spaced across its run, which appears here as logarithmically-spaced spikes in the cost.

### D.3 Analysis of the samples produced by VBMC

We report the results of two control experiments to better understand the performance of VBMC.

For the first control experiment, shown in Fig. S6A, we estimate the log marginal likelihood (LML) using the WSABI-L approximation trained on samples obtained by VBMC (with the $a_{pro}$ acquisition function). The LML error of WSABI-L trained on VBMC samples is lower than WSABI-L alone, showing that VBMC produces higher-quality samples and, given the same samples, a better approximation of the marginal likelihood. The fact that the LML error is still substantially higher in the control than with VBMC alone demonstrates that the error induced by the WSABI-L approximation can be quite large.

For the second control experiment, shown in Fig. S6B, we produce $2 \cdot 10^4$ posterior samples from a GP directly trained on the log joint distribution at the samples produced by VBMC. The quality of this posterior approximation is better than the posterior obtained by other methods, although generally not as good as the variational approximation (in particular, it is much more variable). While it is possible that the posterior approximation via direct GP fit could be improved, for example by using ad-hoc methods to increase the stability of the GP training procedure, this experiment shows that VBMC is able to reliably produce a high-quality variational posterior.

Figure S6: **Control experiments on neuronal model likelihoods. A.** Median absolute error of the LML estimate, as a function of number of likelihood evaluations, for two distinct neurons ($D = 7$). For the control experiment, here we computed the LML with WSABI-L trained on VBMC samples. **B.** Median "Gaussianized" symmetrized KL divergence between the algorithm's posterior and ground truth. For this control experiment, we produced posterior samples from a GP directly trained on the log joint at the samples produced by VBMC. For both metrics, shaded areas are 95% CI of the median, and we consider a desirable threshold to be below one (dashed line). See text for more details, and see also Fig. 3 in the main text.

## Supplementary references

[1] Kingma, D. P. & Welling, M. (2013) Auto-encoding variational Bayes. *Proceedings of the 2nd International Conference on Learning Representations*.

[2] Miller, A. C., Foti, N., & Adams, R. P. (2017) Variational boosting: Iteratively refining posterior approximations. *Proceedings of the 34th International Conference on Machine Learning* **70**, 2420–2429.

[3] Ghahramani, Z. & Rasmussen, C. E. (2002) Bayesian Monte Carlo. *Advances in Neural Information Processing Systems* **15**, 505–512.

[4] Kingma, D. P. & Ba, J. (2014) Adam: A method for stochastic optimization. *Proceedings of the 3rd International Conference on Learning Representations*.

[5] Knuth, D. E. (1992) Two notes on notation. *The American Mathematical Monthly* **99**, 403–422.

[6] Carpenter, B., Gelman, A., Hoffman, M. D., Lee, D., Goodrich, B., Betancourt, M., Brubaker, M., Guo, J., Li, P., & Riddell, A. (2017) Stan: A probabilistic programming language. *Journal of Statistical Software* **76**.

[7] Gelman, A., Carlin, J. B., Stern, H. S., Dunson, D. B., Vehtari, A., & Rubin, D. B. (2013) *Bayesian Data Analysis (3rd edition)*. (CRC Press).

[8] Yao, Y., Vehtari, A., Simpson, D., & Gelman, A. (2018) Yes, but did it work?: Evaluating variational inference. *arXiv preprint arXiv:1802.02538*.

[9] Goris, R. L., Simoncelli, E. P., & Movshon, J. A. (2015) Origin and function of tuning diversity in macaque visual cortex. *Neuron* **88**, 819–831.

[10] Acerbi, L. & Ma, W. J. (2017) Practical Bayesian optimization for model fitting with Bayesian adaptive direct search. *Advances in Neural Information Processing Systems* **30**, 1834–1844.

## Footnotes

[1]This choice of notation makes it easy to apply Gaussian identities used in Bayesian quadrature.