[Reviews · NeurIPS 2018]

Reviewer 1



Summary: The paper considers variational inference in the case where likelihood functions themselves are expensive to evaluate. It suggests approximating the ELBO using probabilistic numerics. A Gaussian process prior is placed on the log joint of the model. A novel acquisition function is proposed along with an approximation of the ELBO for a variational mixture distribution based on the GP posterior and simple Monte Carlo for the mixture entropy. Empirical comparison is performed against a variety of relevant baselines. In meaningful synthetic experiments the proposed method outperforms other methods in higher dimensional cases. In a real example of a neuronal model the proposed models and its variants are the only ones that work. Technical soundness: Overall the paper is technically sound. I would have liked one more compelling real world example. I do also have some questions for the authors which would likely effect my score. Acquisition function: I have some anxiety around the fact that the best performing acquistion function does not depend on the variational parameters (line 159) and that a method that ignores the variational aspect but keeps this search method can often give a better posterior approximation. This seems to run counter to the intuition in line 147 about a sequence of integrals under the corresponding variational approximations. Clearly there are useful things in the algorithm as a whole and the variational marginal likelihood approximation still dominates but should this have changed the emphasis of the paper? Perhaps the (non-variational) search method in equation 8) is the real star? This also relates to some issues around the way the paper is written in the next section. Variational mixture distributions: I have also experienced the difficulties the authors describe (footnote 2) with the mixture bound of Gershman et al 2012. Indeed my general impression was that mixture approximations are hard to get working even with MC approximations. Perhaps this is due to the fact (see Jaakkola and Jordan 1998 which the authors shouldd cite) that the improvement in ELBO is at best logarithmic in the number of mixture components and can therefore be lost beneath the Monte Carlo noise. Did the authors find that adding mixture components beyond two or three gave much improvement? Did they rule out that this was an optimization effect? Were the results sensitive to initialization? Annealed importance sampling: I am surprised by how poorly AIS performed. My understanding from looking at the Gunter et al paper is that the proposal distribution used was a Metropolis-Hastings one. Did the authors look at multivariate slice sampling? This would likely be less sensitive to step sizes and indeed the authors proposed method calls slice sampling as a subroutine. Clarity: Apart from the questions discussed above the paper is well written. It references key literature. It is measured and resists hyperbole, carefully drawing the readers attention to potential limitations. I do think that an expert would have a good chance of reproducing the key results given what is written. I would have liked more discussion of equation 8. This seems to be a key point in the paper and has not received sufficient emphasis. I would have really liked a visualisation of the algorithm in action in a case that was simple enough to plot. This would have helped answer my questions about the variatonal approximation above. It is great that the authors are willing and able to release code after the review process. Why not release it in an anonymized repository at this stage next time? This would help build reviewer confidence in the submission. I suggest that the authors include larger versions of Figures 1 and 2 in the supplement. The references need tidying up. For instance in many cases the publication venue has been ommitted. Originality and significance: There is an interesting new idea here which I think could be important going forward. Specifically, I like the idea that probabilistic numerics and variational inference can be usefully combined. Given my concerns around the non-variational acquisition function I'm not sure it has been fully exploited yet. The solid experimental contribution should also be given credit. References: Gunter, Tom and Osborne, Michael A and Garnett, Roman and Hennig, Philipp and Roberts, Stephen J (2014), Sampling for Inference in Probabilistic Models with Fast Bayesian Quadrature, Advances in Neural Information Processing Systems 27 Gershman, S., Hoffman, M., & Blei, D. (2012) Nonparametric variational inference. International Conference on Machine Learning. Tommi S. Jaakkola , Michael I. Jordan (1998) Improving the Mean Field Approximation via the Use of Mixture Distributions. Learning in Graphical Models. Post author feedback comments: I have read the author response. I am satisfied with their reply and have increased my score.

Reviewer 2



Summary The authors propose a sample-efficient framework for variational inference for low-dimensional (D<=10) problems in which evaluation of the joint is computationally expensive. This framework approximates the log joint with a GP surrogate and employs a variational posterior in the form of Gaussian mixture. Given this choice of variational distribution, the posterior mean and variance of the expected value of the log joint can computed analytically. The authors now iterate between an active sampling step that improves the GP approximation of the log joint, and updates to the variational distribution via stochastic gradient descent. The authors evaluate their framework on three families of synthetic functions and a computational model of neuronal activity in the V1 and V2 regions of the visual cortex in macaques. Quality, Originality, and Significance This is a fairly well-written and overall reasonable submission. Combining Bayesian quadrature with variational inference seems like a nice (if inevitable) idea, and the authors execute on this idea in a manner that results in a fairly straightforward framework, which in my opinion is a plus. The main thing that I have trouble with in this paper is identifying when VBMC is the right tool for the job. The paper would be much stronger if the authors were able to come up with a compelling motivation for the problem at hand. In other words, what is the set of problems for which we (a) have a distribution with a low-dimensional prior and (b) a really expensive likelihood, for which (c) we would really like to obtain an approximation to the marginal likelihood and the posterior, and (d) where we can reliably learn a GP surrogate in <1000 samples (at which point GP scalability starts to become an issue). I also found it difficult to evaluate the significance of the experimental results (owing in part to the lack of detail in the description). Clarity The paper spends a fair amount of time establishing notation for all the background components of the proposed work (varitioanal inference, bayesian quadrature), which I appreciate. Unfortunately, the discussion of the experiments ends up being fairly terse. It's quite hard to understand exactly what problems are being considered in the experiments. For the the synthetic data, 3 families of functions are described in a long paragraph that contains the relevant information, but could be easier to parse. For the experimental neuronal recording from the macaque V1 and V2 cortex, it would be helpful to have a more self-contained explanation of the problem at hand. The authors cite a reference in Neuron [12], which I presume very few readers will be familiar with. They then reference "auxilliary parameters" and transformation of constrained parameters via shifted and scaled logit transform, but it is very unclear what is going on. I of course appreciate the challenges of fitting details in the NIPS page format, but at the very least the authors could provide additional details and discussion in the supplementary material? Minor - How is q^*(x) a "proposal" in proposal uncertainty sampling? Can this be interpreted as some form of importance sampling? - For both dataset -> For both datasets - likelihoood -> likelihood (please run an extra spell check) - approximated via -> approximated with/by Post-Response Comments I've read the author response, which overall is reasonable. I am inclined to agree with the point that the ability of VBMC to provide uncertainty estimates in cases where we would otherwise perform Bayesian optimization is interesting. I was a little disappointed by the assertion that the the space of problems in which VBMC applies "is quite large and of outstanding interest in fields such as computational neuroscience". While the authors provide a citation, I would have appreciated more of an actual explanation or concrete examples. That said, I'm inclined to say that this is just about above the bar for acceptance, and have adjusted my score upward.

Reviewer 3



[Post-response update] I have read and am satisfied with the author response. My score has been adjusted. [Original review] Paper overview: - The authors propose to combine variational inference with Bayesian Monte Carlo to perform joint inference of the posterior and evaluation of the log marginal likelihood. The use case is expensive-to-evaluate likelihood functions. Active sampling is used to select query points for model parameters. Experiments demonstrate the effectiveness of the approach over recent BQ-type methods. Review summary: - The idea of combining VI and BMC is interesting, and the experiments show that the proposed method improves upon recent methods. There is some intuitive exposition missing from the paper (example below). Adding would help in understanding the significance of the contribution. The experiments are extensive which dampens some of the concern arising from the heuristics involved in the method. Unfortunately, there is no discussion or even comment on the computational considerations involved for the proposed method. Some discussion on the methods' limitations would have been nice. Clarity: - The algorithm and experimental procedure are generally well described. But, there seems to be a lack of intuitive explanation for design choices. For example, if the joint posterior is modeled as a GP with unimodal mean function, why is the variational approximation multi-modal? - Why do BMC and WSABI-L share the same color coding in Figures 1 and 2? Originality: - The idea of combining variational inference and Bayesian Monte Carlo is novel as far as I know. Quality: - The experimental procedure is extensive in consideration of likelihood types. However, comparison wrt/ the real-world datasets of Gunter et al. (2014) would have been nice. Also, since there is no discussion of computational considerations for the proposed method, performance wrt/ wall-clock time would have been preferred. - Since the symmetrized KL is computed with moment-matched Gaussians, it is not clear how valuable a metric it is in assessing the posterior approximation quality. - Can the authors speak to the performance of the proposed method beyond D=10? - What are the weight distributions of the variational posterior mixture components like at convergence/stopping? In the synthetic experiments, how do the mixture components compare to the real components? How does the method perform with a fixed no. of components, e.g., 1,5? - There are some heuristics involved in the algorithm - increase of mixture components w/ no. of samples, no. of active points to sample, no. of GP samples to perform approximate hyperparameter marginalization, definition of plausible regions - which create some concern regarding the generalization performance of the method.